behaviour, physiology, evolution

insect migration, orientation, time-compensated sun compass, hoverfly, flight simulator, navigation

**Author for correspondence:**
Karl R. Wotton
e-mail: k.r.wotton@exeter.ac.uk

# Hoverflies use a time-compensated sun compass to orientate during autumn migration

Richard Massy[1], Will L. S. Hawkes[1], Toby Doyle[1], Jolyon Troscianko[1], Myles H. M. Menz[3,4,5], Nicholas W. Roberts[6], Jason W. Chapman[1,2,7] and Karl R. Wotton[1]

[1]Centre for Ecology and Conservation, and [2]Environment and Sustainability Institute, University of Exeter, Cornwall Campus, Penryn, UK
[3]Department of Migration, Max Planck Institute of Animal Behaviour, Radolfzell, Germany
[4]Department of Biology, University of Konstanz, Konstanz, Germany
[5]School of Biological Sciences, The University of Western Australia, Crawley, WA, Australia
[6]School of Biological Sciences, University of Bristol, Bristol BS8 1TQ, UK
[7]Department of Entomology, Nanjing Agricultural University, Nanjing, People's Republic of China

MHMM, 0000-0002-3347-5411; NWR, 0000-0002-4540-6683; JWC, 0000-0002-7475-4441; KRW, 0000-0002-8672-9948

The sun is the most reliable celestial cue for orientation available to daytime migrants. It is widely assumed that diurnal migratory insects use a 'time-compensated sun compass' to adjust for the changing position of the sun throughout the day, as demonstrated in some butterfly species. The mechanisms used by other groups of diurnal insect migrants remain to be elucidated. Migratory species of hoverflies (Diptera: Syrphidae) are one of the most abundant and beneficial groups of diurnal migrants, providing multiple ecosystem services and undergoing directed seasonal movements throughout much of the temperate zone. To identify the hoverfly navigational strategy, a flight simulator was used to measure orientation responses of the hoverflies *Scaeva pyrastri* and *Scaeva selenitica* to celestial cues during their autumn migration. Hoverflies oriented southwards when they could see the sun and shifted this orientation westward following a 6 h advance of their circadian clocks. Our results demonstrate the use of a time-compensated sun compass as the primary navigational mechanism, consistent with field observations that hoverfly migration occurs predominately under clear and sunny conditions.

## 1. Background

Migratory insects dominate aerial bioflows in terms of diversity, abundance and biomass [1]. They provide key ecosystem services, connect distant environments and have profound consequences for the functioning of ecosystems [2–4]. Entomological radar studies over the southern United Kingdom suggest that daytime migrants make up greater than 70% of total insect numbers [5]. The larger insects in this group (body mass greater than 10 mg), which include hoverflies, ladybirds, carabid beetles and butterflies, undergo seasonally directed migrations [5,6]. Importantly, while those insects making directed migrations form only a small percentage of total numbers (less than 1%), they make up a sizable percentage of total biomass (approx. 20%), underlining their ecological importance within the migratory assemblage [5]. While seasonally beneficial migration directions have been observed in many insect groups [7–11], the precise mechanisms used for orientation during diurnal migration have only been identified in a few butterfly species: the eastern North American population of the monarch butterfly (*Danaus plexippus*), and in the neotropical butterflies *Aphrissa statira* and

*Phoebis argante* [12,13]. To travel in their seasonably favourable direction, individuals undergo menotaxis according to the position of the sun, which requires compensating for the changing position of the sun throughout the day. This is known as a 'time-compensated sun compass' [12,13]. The key test used to infer the presence of time compensation involves the clock-shift procedure whereby an animal's internal circadian clock is shifted out of phase of normal daylight hours. This causes the animal to predictably misinterpret the direction of the sun, orientating instead based upon their shifted internal time [14]. The neural mechanism behind the time-compensated sun compass is derived from an interaction between the circadian clock and the central complex of the brain [15,16], with evidence in monarch butterflies suggesting that the process is entrained by synchronizers in the antennae [17,18].

Tethered flight experiments in *Drosophila* have demonstrated that the sun and patterns of polarized light in the sky can be used to maintain a constant heading [19,20]. However, *Drosophila* adopt arbitrary headings with respect to a simulated sun with no evidence of time compensation [19]. While such path straightening mechanisms may be sufficient for shorter flights (for example up to 10 km over a few hours seen in *Drosophila* [20]), the longer distance and duration of migratory flight, as seen in some other Dipterans, requires a method for orientating towards a seasonally favourable heading, while potentially also compensating for the movement of the sun through the day. Numerous other species of Diptera are known to undertake longer directed flights, including several species in the family Syrphidae (hoverflies), where during migration, flight often persists throughout daylight hours and over hundreds of kilometres [21,22]. Hoverfly long-distance migratory behaviour appears to be widespread within temperate zones [6,23–27] and results in the movement of huge numbers of individuals. For example, it has been estimated that up to four billion *Episyrphus balteatus* (marmalade hoverfly) and *Eupeodes corollae* (vagrant hoverfly) travel over the southern region of Britain annually [6]. In addition, hoverfly migration is likely to be geographically widespread with important ecological impacts in terms of pollination, the control of pest aphids, the breakdown of organic matter, nutrient transfer and the structuring of food webs [2,6,28].

Patterns of hoverfly migration are best understood in Europe where seasonal influxes into northern regions begin in spring and are followed by a broad front southward migration during autumn [6,22,23,29]. Numerous observations, together with stable isotope analysis, suggest that individuals leaving northern Europe may travel thousands of kilometres to potential overwintering sites around the Mediterranean basin and North Africa [30,31]. This appears to be achieved by a combination of high-altitude wind-assisted flight and, when facing headwinds, low-level flight within the flight boundary layer (FBL) [32,33]. Flying within the FBL lessens the impact of headwinds as wind speed is lowest at ground level and migration is most easily observed along coasts or through mountain passes where the topology serves to concentrate migrant numbers: for example in the mountain pass of Bujaruelo in the Pyrenees [29]. Work by Aubert *et al.* in the 1960s made use of these low-level flights to carry out mark-recapture experiments on insects during the autumn migration period in the Swiss Alps [21]. These experiments showed that in good conditions (sunny and light winds), migratory hoverflies including *Eristalis tenax*, *Ep. balteatus* and *Syrphus vitripennis* were able to cover 3 km

in 10–15 min (12 to 18 km h$^{-1}$). The most distant recoveries were made 111 km away with a moderate tailwind of 6–8 m s$^{-1}$ and the authors suggest that hoverflies covered this distance in a single flight [21]. More recently, radar studies of high-altitude hoverfly migration indicate average speeds of autumn migration of around 10 m s$^{-1}$ (36 km h$^{-1}$) aided by the selection of favourable winds [6,34], suggesting the proposed migratory distances are indeed achievable over a relatively short period of time. In addition to estimates of flight speed, radar studies have revealed sophisticated strategies used by migratory hoverflies, including the ability to select favourable winds, to partially compensate for wind drift and to orientate towards seasonally preferred directions, particularly during autumn migration [35,36]. Mean directions of hourly tracks and headings from these radar studies displayed no difference throughout the entire day, suggesting the presence of a compensation mechanism for the sun's changing azimuth [34].

To investigate whether hoverflies use a time-compensated sun compass to orientate, we undertook a series of tethered flight experiments on actively migrating *Scaeva pyrastri* (pied hoverfly) and *Scaeva selenitica* (yellow-clubbed hoverfly), caught at ground level during their southward journey over the Pyrenees. We predict that hoverflies would continue to orientate southwards in a seasonally favourable direction when given a view of the sky including the sun but excluding geographical features, and that when subjected to the clock-shift procedure, hoverflies would predictably shift their orientation in accordance with the solar azimuth at their perceived time.

## 2. Methods

### (a) Migrant collection

Initial investigation of flight behaviour in a range of species of actively migrating hoverflies was carried out in Switzerland from August to October 2017 and revealed *S. selenitica* and *S. pyrastri* to be the most consistent fliers during tethered flight experiments. We therefore selected these species for further investigation, hereafter referring to them as 'hoverflies'. During September and October of 2018 and 2019, migrating hoverflies were caught using hand nets as they traversed the Pyrenees, 2273 m above sea level at the mountain pass of Bujaruelo on the French–Spanish border (42.7038793 N, −0.0641454 W; figure 1*a*). Permission to conduct experiments was obtained from the Parc national des Pyrénées (France, authorization numbers: 2018-9 and 2019-67) and the Gobierno de Aragon (Spain, authorization numbers: 500201/24/2018/06141 and 500201/24/2019/02174). Flies were housed in 30 cm cubed mesh cages (Watkins & Doncaster) and flown immediately or transferred to the village of Gavarnie for further experiments. Flies that were stored for later experiments were kept outside to maintain their circadian rhythm to natural daylight hours and supplied with 40% honey solution, pollen and water ad libitum.

### (b) Flight simulator

To study the hoverflies response to celestial cues, flight direction was measured using a portable flight simulator with magnetic tethering (figure 1*b*). We constructed the flight simulator from a white 200 mm diameter × 250 mm high PVC cylinder, commonly used as ducting. This contained a large Neodymium ring magnet (K&J Magnetics, Inc. no. RX8CC, 38 mm outside diameter × 19 mm inside diameter × 18 mm thick). The magnet was

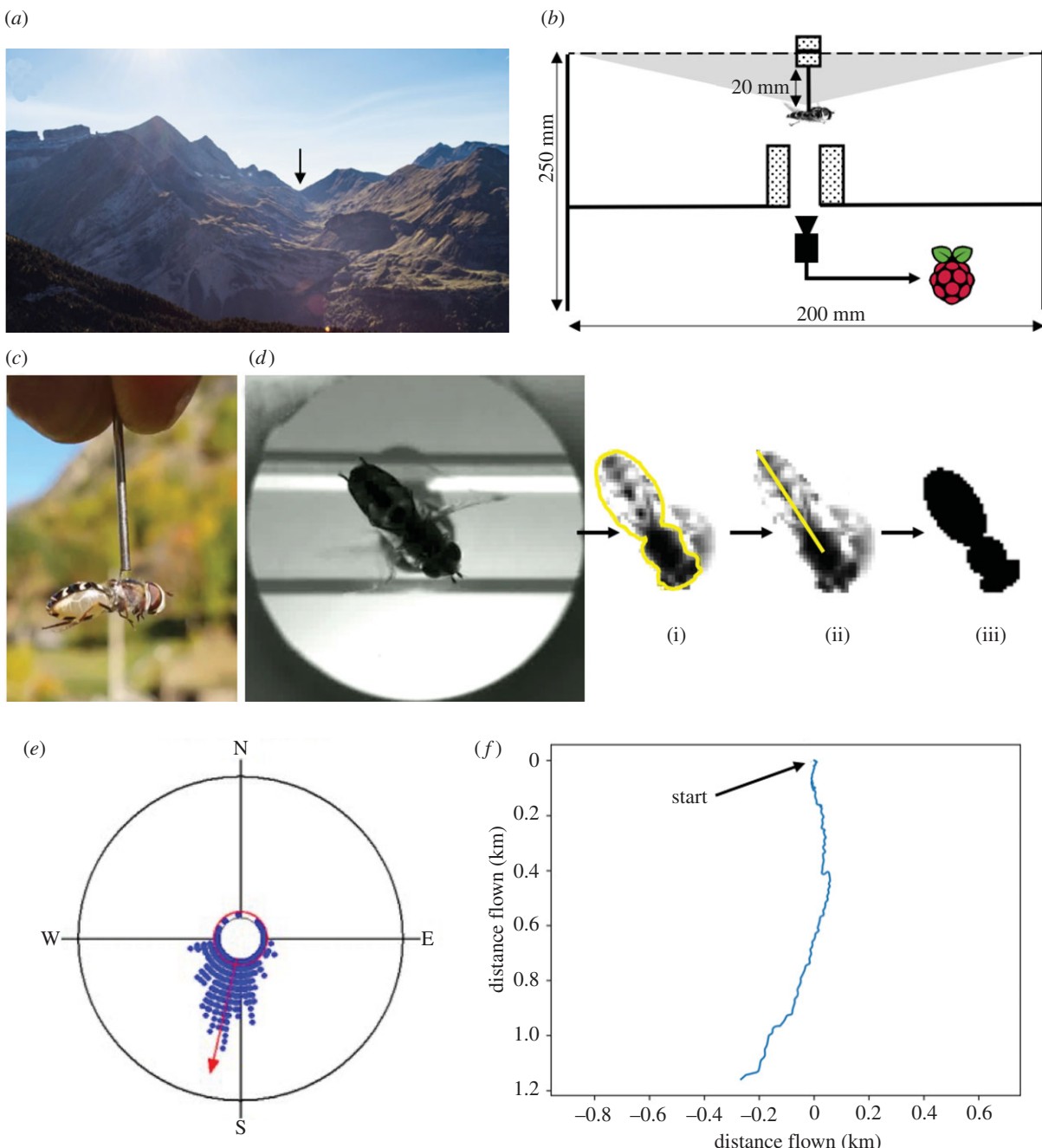

**Figure 1.** Hoverfly flight simulator. (*a*) Active migrants were collected from the Puerto de Bujaruelo (black arrow), a 2273 m pass on the French–Spanish border in the Pyrenees. (*b*) Flight simulator schematics and visual field presented to the hoverfly (light grey: full visual angle 116°). (*c*) A tethered *Scaeva pyrastri*. (*d*) A still taken from a flight simulator video and representative data extraction steps: after loading the stack of photos corresponding to an experiment, contrast is increased for all photos by decreasing maximum levels until background detail disappears; (i) the hoverfly body is outlined on one frame by manually drawing an ellipse over the abdomen and a circle over the thorax, then unifying the shapes; (ii) the resultant angle between thorax and abdomen is checked by eye; (iii) this outline creates a kernel which is rotated for each image to find the angle of best fit. (*e*) Circular histogram displaying the data from a 5 min experiment of a single hoverfly. Each blue dot represents three recorded angles, the red arrow indicates overall mean direction while the length of the arrow indicates *r*, the measure of flight directedness, ranging from 0 (random distribution) to a maximum of 1 at the edge of the circle. (*f*) Virtual flight path for the hoverfly observed in (*e*). Axes show distance travelled in kilometres under a constant 5 ms$^{-1}$ flight speed. Flight speed is estimated from mark-recapture experiments on migratory hoverflies carried out by Aubert *et al.* between the Col de Bretolet on the Switzerland–France border and the Col de la Golèse in France [21]. Hoverflies frequently completed the 3 km distance in 10–15 min indicating a minimum flight speed of 3–5 ms$^{-1}$. (Online version in colour.)

supported centrally on a transparent acrylic shelf above a 20 mm central hole. A raspberry Pi camera (Pi v. 2.1 8MP 1080p camera module) mounted to the underside of the shelf filmed flight behaviour from below. The image was manually focused by screwing the lens fitting. A button attached to the raspberry Pi (Pi 3 Model B running Raspbian) initiated a custom Python script (electronic supplementary material, file S1) that recorded flight behaviour for 5 min, with recording status indicated by an illuminated LED. A transparent acrylic strip (20 mm wide) resting

on top of the flight simulator supported two 15 × 6 mm neodymium magnets (Magnet Expert Ltd), one above and one below, positioned directly above the lower ring magnet. Visual cues were obscured in the flight simulator with a lid that allowed a 116° visual field of the sky and sun but obscured the surrounding landscape. In preliminary experiments, hoverflies were found to orientate directly towards sunlight illuminating the opposite wall inside the flight simulator. To prevent this, a layer of the photographic diffuser (quarter white, LEE filters) was used to

cover the opening which diffused the light more equally within the flight simulator (although reducing the level of polarization by 50% and transmitting 80% of overall light). The reduction in the degree of polarization was calculated from Stokes' parameter measurements, using a Glan Thompson polarizer, of the light both before and after passing through the diffuser [37].

## (c) Experimentation

Sun compass experiments were undertaken in the flight simulator between 10.37 and 16.34 local time. Immediately prior to experimentation, flies were restrained on a sponge using a weighted mesh and glued by the thorax to a truncated sewing needle (steel, diameter $1.0 \times 20$ mm) with ultraviolet curing cement (Bondic™; figure 1c). Tethered hoverflies were then placed inside the flight simulator with the needle touching the upper magnet, where they were held in place but free to rotate in the horizontal plane. The video recording was then initiated, filming the hoverfly for 5 min.

To exclude the sun but still include intensity and polarized-light gradient cues, a lid with a smaller circular opening was used to restrict the field of view to 70°. These experiments were undertaken in the evenings between 16.17 and 18.45 when levels of overhead polarization were greatest and, as the sun had passed behind the mountains, without the photographic diffuser. Flies used in this experiment came from the same stock as the sun compass experiment which were exposed to normal daylight hours.

Finally, to investigate time compensation, a separate batch of flies was subjected to a 6 h advanced clock shift according to the mountain daylight time (lights on 6 h after sunrise, off 6 h after sunset). To achieve this, hoverflies were stored for 10 days in a larger $90 \times 60 \times 60$ cm mesh cage (Watkins & Doncaster), blacked out and illuminated on a timer by a 7.5 W, 4000 k colour-temperature LED bulb. Hoverflies were fed ad libitum as before, but owing to larger cage dimensions, food and water was additionally attached to the sides of the cage and branches were added to provide shelter. The experiments were undertaken with the same methodology as the sun compass, but later in the day to allow a minimum of 1 h after the artificial sunrise before experimentation began. The difference in the sun's azimuth between the time of the experiment and the insect's expected time, 6 h earlier, ranged from 91.1° to 104.5° depending upon the date and time of the experiment (mean = 98.1°). With full time compensation for the sun position, flies would be expected to alter their heading by an angle within this range. Experiments were carried out between 7 and 11 October 2019 between 14.29 and 16.42, overlapping with a subset of the sun compass and restrictor experiments.

## (d) Data analysis

Analysis of the videos of hoverfly flight was first undertaken by hand, noting the timings of the initiation and cessation of flight as well as any perturbations (such as a shadow or the fly spinning out of control). Only recordings that included 70 s or more of uninterrupted flight were selected for analysis, with the first 10 s discarded to allow time for acclimatization and orientation. Video files were converted into jpg images at a frequency of 5 Hz (5 frames s$^{-1}$) using TOTAL VIDEO AUDIO CONVERTER [38]. Body orientation was then calculated from these images using an automated kernel-matching process in IMAGEJ [39]. The script requires the user to draw outline ellipses around the head, thorax and abdomen of the hoverfly in the first frame to create a kernel, and then automatically tracked the angle of the body in subsequent frames to the nearest 1 degree using a convolution filter (figure 1d; electronic supplementary material, file S2 for IMAGEJ script and file S4 for raw outputs).

Clock-shifted experiments were undertaken later in the day, so to control for the naturally more clockwise position of the sun (i.e. to rule out simple phototaxis) angles were rectified to be relative to the sun's azimuth so that the sun was in the 180° position. To achieve this, for each recording, every angle was rotated by the difference between geographical south and the sun's azimuth when the experiment was undertaken. To enable calculation of the angle difference, nonlinear least-squares models were fitted using the dose–response model (function L.4) from the R package drc (v. 3.0-1) [40] to azimuth-time data for each day of experimentation (azimuth data specific to the field site was obtained from sunearthtools.com). These models enabled the prediction of the azimuth at the time of every experiment, after which angles were adjusted by the difference between 180° and the azimuth.

Statistical analysis and graphing were carried out in R v. 3.5 (R Core Team, 2020) using R STUDIO 1.3 and the packages circular (0.4-93) [41], plotrix (3.7-8) [42] and glmmTMB (1.0-1) [43]. To test for differences between sexes or species, generalized beta-family mixed-models from the R package glmmTMB were used with experiment type (sun compass or time-shifted sun compass) as the random term to account for potential behavioural differences between experiments. A custom R-script was made for the Moore's Rayleigh test, which to our knowledge does not currently exist in an open-source format. This non-parametric test is an analogue to the Rayleigh test, used for weighted data [44,45]. The test statistic: $R^*$ is an arbitrarily scaled alternative to $r$ of circular regression. This is compared to a statistical table of probabilities to test the null hypothesis. The corresponding rank-weighted mean direction is also given, for which an additional script was created to bootstrap confidence intervals. All custom scripts are available in the electronic supplementary material, file S3.

## (e) Migrant activity patterns and temperature

The activity pattern of dipteran migrants crossing the mountain pass of Bujaruelo in 2019 between 6 September and 11 October was extracted from video footage. A video camera sampled a $2 \times 2$ m cross-section of the mountain pass for 1 min every 15 min between 9.00 and 17.00. The number of individuals crossing the midpoint of the frame southwards were counted manually for each timepoint and averaged across all the recorded days. Temperatures were taken from a temperature logger located on the pass.

## 3. Results

Hoverflies flown in the sun compass experiment, with the sun visible, landscape cues obscured and other celestial cues attenuated by photographic diffuser, headed almost due south (Moore's modified Rayleigh (MMR) test: $\theta = 188.2°$, $n = 30$, $R^* = 1.294$, $p < 0.01$; figure 2a). Diurnal activity patterns of migrants crossing the mountain pass show a marked increase beginning after 14.00, coinciding with increasing temperatures (figure 2b). Splitting the data of the sun compass experiment in half, hoverflies flown before 13.57 orientated randomly (MMR test: $R^* = 0.375$, $n = 15$, $p < 0.9$; figure 2c), whereas hoverflies flown after this time, hereafter referred to as the 'last 15', displayed a more directed group heading to the south-southwest (MMR test: $\theta = 194.4°$, $n = 15$, $R^* = 1.480$, $p < 0.0001$; figure 2d). Traditional Rayleigh tests produced comparable results (table 1).

To explore the role of celestial cues other than the sun, hoverflies were flown with a restricted view of the sky, which excluded the sun but provided an unobscured view

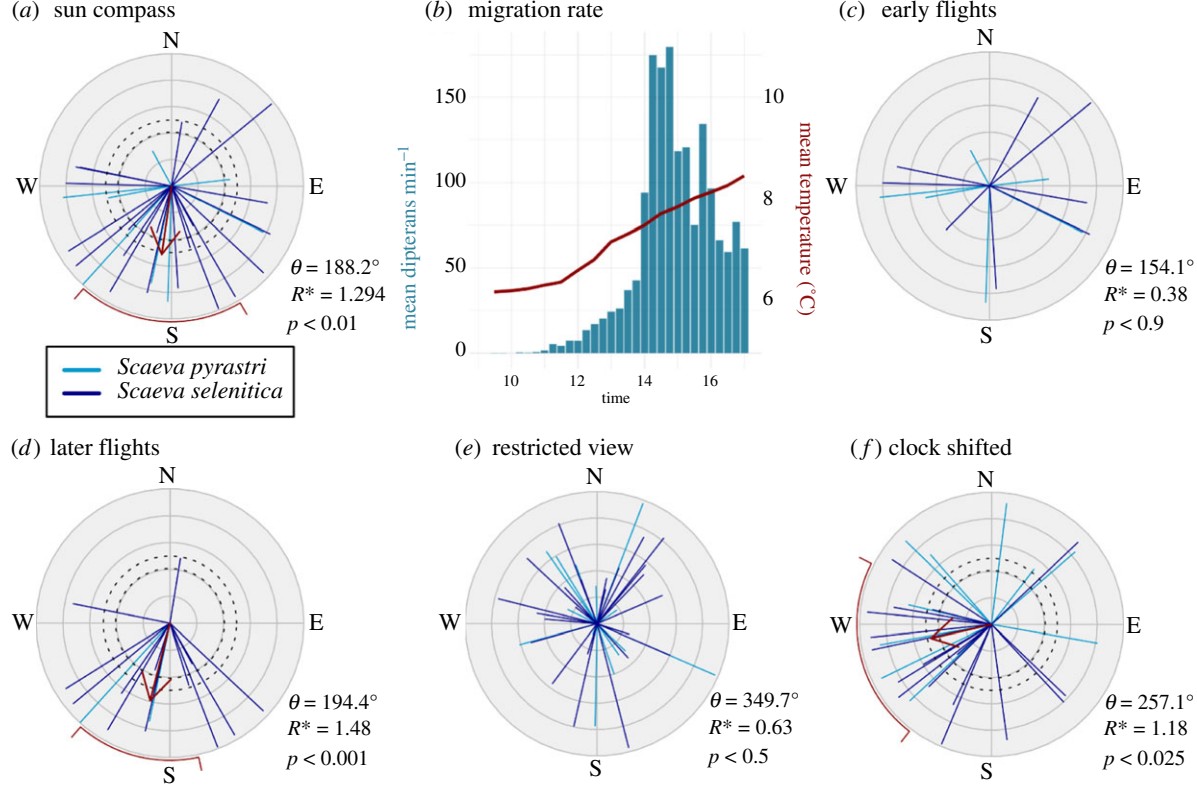

**Figure 2.** Hoverfly orientation in a flight simulator under various conditions and diurnal activity patterns. (a) Hoverfly orientation in a flight simulator under clear-sky conditions. Blue lines indicate the vectors of individual hoverflies with colour indicating the species and length corresponding to the $r$ value ($r = 1$ at the outer radius). Red arrows indicate the group weighted mean direction with length depicting $R^*$ relative to 2.5 at the outer radius. Dashed circles represent the significance intervals of 95% (inner) and 99% (outer). Red bars outside of the circle indicate 95% weighted confidence intervals. (b) Diurnal activity pattern of hoverfly migrants as a rate of southbound movement over the mountain pass and mean temperature. (c,d) the results of (a) split by time of day: (c) hoverflies flown before 13.53 (first 15) and (d) hoverflies flown after 13.57 (last 15). (e) Under restricted views conditions with the sun obscured but light intensity, chromatic and polarization patterns visible. (f) Hoverflies that had undergone a 6 h advanced clock shift. Numerical results are presented in table 1. (Online version in colour.)

of light intensity, chromatic gradients and polarized light. These hoverflies orientated randomly (Moore's Rayleigh test: $R^* = 0.634$, $n = 42$, $p < 0.5$; figure 2e). To account for the ambiguous nature of the polarized-light cue, where opposite directions along an axis cannot be distinguished, we tested for a bidirectional line-up response by bidirectionally transforming the angles and repeating the test; however, the results remained non-significant (Moore's Rayleigh test: $R^* = 0.585$, $n = 42$, $p < 0.5$).

Hoverflies exposed to a 6 h advanced clock-shift treatment shifted their heading westwards ($\theta = 257.2°$, $n = 27$, $R^* = 1.177$, $p < 0.025$; figure 2f). A Mardia-Watson-Wheeler test confirmed that the mean directions of hoverflies under clock-shifted conditions were significantly different from the sun compass group both in its entirety ($W = 6.50$, $p = 0.039$) and for the last 15 flown ($W = 9.43$, $p = 0.0089$). The mean direction of the time-shift experiment was 68.9° clockwise from that of the sun compass experiment and 62.7° as compared to the last 15, representing 64–70% of the 98.1° adjustment required to fully compensate for the mean 6 h azimuth change and 70–77% of the adjustment required under a linear compensation scenario of 15° h$^{-1}$. However, confidence intervals (table 1) do not rule out complete time compensation.

As the clock-shifted experiments were undertaken later in the day when the sun would naturally be in a more clockwise position, to rule out simple phototaxis, the analyses were repeated on angles that were rectified to be relative to the direction of the sun, so that the direction of the sun (instead of south) is in the 180° position (electronic supplementary material, figure S1). The rectified mean directions of hoverflies under normal and clock-shifted conditions still differed significantly, as confirmed by a Mardia-Watson-Wheeler test (all: $W = 7.75$, $p = 0.021$; last 15: $W = 10.918$, $p = 0.0043$), while back transforming the angle of the clock-shifted hoverflies based on their perceived time results in a group heading that is not significantly different from the sun compass (all: $W = 2.76$, $p = 0.251$; last 15: $W = 4.06$, $p = 0.132$). Together this indicates that the hoverflies were indeed clock-shifted, flying in different directions with respect to the sun and compensating for its position rather than just following its course throughout the day.

Mixed sexes and species of *Scaeva* hoverflies were analysed in this study. Of 99 hoverflies analysed, 65 were *S. selenitica* (62 females and three males) and 34 were *S. pyrastri* (30 females and four males), with sex and species ratios representative of the capture rate in the field. To explore whether there were behavioural differences between sexes or species, we pooled the vector data from the different experiments together and investigated two parameters of the recorded flights: flight directedness ($r$-values) and deviation from the experimental mean direction (as calculated from the Moore's Rayleigh test). There was no significant effect of sex ($p = 0.477$) or species ($p = 0.657$) on flight directedness ($r$), and a lower relative model fit than the intercept only model Akaike information criterion (AIC) (AIC$_{sex+species}$ −32.6, AIC$_{intercept\ only}$ −35.8). To investigate deviation from the experimental

**Table 1.** MMR test and parametric circular analysis (Rayleigh test) of the mean directions of each experiment. (The angles from rectified experiments have been adjusted to be relative to the sun's azimuth (azimuth = 180°). Confidence intervals bootstrapped from $10^6$ resamples. Italicized mean directions calculated from bidirectionally transformed angles on a scale of 0°–180°.)

| | n | mean direction: weighted and (non-weighted) | 95% confidence interval: weighted and (non-weighted) | R*: MMR | r-value: Rayleigh | p: MMR & (Rayleigh) |
|---|---|---|---|---|---|---|
| sun compass | 30 | 188.2° (188.7°) | 149.4°–220.9° (148.7°–226.4°) | 1.294 | 0.384 | <0.01 (0.011) |
| sun compass first 15 | 15 | 154.1° (155.3°) | — | 0.375 | 0.130 | <0.9 (0.782) |
| sun compass last 15 | 15 | 194.4° (194.8°) | 167.7°–220.9° (171.0°–221.0°) | 1.480 | 0.664 | <0.001 (0.0007) |
| restricted view | 42 | 349.7° (340.9°) | — | 0.634 | 0.208 | <0.5 (0.208) |
| restricted view (bidirectional) | 42 | *166.9° (163.6°)* | — | 0.585 | 0.177 | <0.5 (0.270) |
| clock shift | 27 | 257.1° (253.5°) | 217.6°–296.8° (218.9°–290.2°) | 1.177 | 0.416 | <0.025 (0.008) |
| sun compass rectified | 30 | 173.3° (173.5°) | 142.3°–203.3° (141.0°–211.4°) | 1.433 | 0.408 | <0.005 (0.006) |
| sun compass last 15 rectified | 15 | 170.4° (168.4) | 146.4°–192.7° (146.2°–191.8) | 1.549 | 0.6873 | <0.001 (0.0004) |
| clock shift rectified | 27 | 232.4° (228.2°) | 198.1°–275.1° (194.8°–264.2°) | 1.196 | 0.426 | <0.025 (0.007) |

mean direction, the potential deviation of −180° to +180° was normalized to 0–1 by dividing by 180 and changing negatives to their additive inverse. For this, only experiments with significant mean directions were used (sun compass and clock-shifted sun compass). There was also no significant effect of sex ($p = 0.461$) or species ($p = 0.593$) on deviation from the experimental mean direction and a lower relative model fit than the intercept only model ($AIC_{sex+species}$ −6.12, $AIC_{intercept\ only}$ −9.41). See the electronic supplementary material, table S1 for effect estimates.

A Mardia-Watson-Wheeler test confirmed the lack of significant directional bias by species of *S. selenitica* and *S. pyrastri* for the time-shift experiment ($W = 3.7845$, $p = 0.1507$). As 10 individuals in each group are required for the chi-squared comparison, insufficient *S. pyrastri* ($n = 9$) prevented a comparison for the sun compass experiment. Low numbers of males prevented an inter-sex comparison; however, males had on average a lower deviation from the group mean direction than females, making excess leverage owing to male outliers unlikely. For these reasons, we are satisfied that any behavioural differences between sexes and among species are minor enough to justify pooling the data.

## 4. Discussion

The sun is thought to be the most important cue for diurnal migrants. Here, we demonstrate the ability of *Scaeva* spp. hoverflies to orientate in a southerly direction during autumn migration when given a view of the sky including the sun but excluding geographical features. This result is supported by previous radar studies showing southward migration headings of *Ep. balteatus* and *Eu. corollae* hoverflies over southern Britain in the autumn (reporting a 180° track and 198° heading [34] versus a 188° (all) or 194° (last 15) heading in this study). This gives us confidence that the directional flight observed in the flight simulator is equivalent to migratory directions in the field. This ability to orientate south is lost when hoverflies are given a restricted view of the sky that excludes the sun, indicating that the sun acts as

the primary celestial cue. Spring and autumn migratory *Ep. balteatus* and *Eu. corollae* hoverflies have been shown to orientate differently, with autumn migrants displaying stronger orientation tendencies than spring migrants [34]. These studies revealed high group directedness in autumn hoverfly migrants, with an *r*-value of 0.78 which is comparable to the group directedness measured for autumn migratory monarch butterflies in flight simulator experiments (*r*-values of 0.83 in [13] and 0.8 in [46]). We observed lower directedness in hoverflies flown earlier in the day ($r = 0.13$) but high directness later in the day for the last 15 flown ($r = 0.66$). Despite this, these values remain lower than in other systems and may represent a less directed strategy employed by autumn migrating *Scaeva* spp. hoverflies, experimental noise, or a combination of both factors. Our methodology differed from butterfly experiments in the use of a magnetic tethering system, attenuation of celestial cues with a diffuser layer and a lack of laminar airflow that may have each contributed to poorer orientation responses. We note however that monarch butterflies navigate to a specific location during their autumn migration, so might be more committed to maintaining the appropriate direction when flown in a flight simulator. Hoverflies migrating to warmer climates without a specific destination could be more flexible and adapt their flight headings more readily to the prevailing conditions. This might result in a lower concentration of observed directions as alternative vectors might come at very little cost to fitness.

We observed a lack of directness in flights undertaken earlier in the day that may be directly controlled by circadian behavioural patterns such as early day feeding or indirectly by the effect of low temperatures on activity. Temperature appears to be an important motivating factor for migratory flight in Monarch flight simulator experiments [13] and rates of hoverfly migration on the pass only show large increases at 14.00, around solar noon, as temperatures build. In support of temperature as a factor, we note that at Randecker Maar (Germany) at 773 m above sea level, *S. pyrastri* migration typically reaches its peak earlier in the season than in the Pyrenees, and with most individuals active between 10.00 and 16.00 in relatively even numbers.

As migration is a physiologically demanding activity, it may be that in the Pyrenees, hoverflies optimize their flight time to be later in the day, which would not be necessary at lower elevations and earlier in the season owing to the warmer temperatures. The roles of temperature, photoperiod and solar elevation warrant further investigation into their potential roles in modifying migratory flight times.

As has been demonstrated in migrating butterflies [12,13], our time-shift experiment shows that migratory hoverflies compensate for the position of the sun as it changes throughout the day. The degree of compensation observed in our experiments was approximately two-thirds of the average change in azimuth, although confidence intervals do not rule out complete compensation as compared to the mean direction of the sun compass experiment. Similar results have been documented for neotropical migrant butterflies while orientations of monarch butterflies more closely match predicted shifts [12,13]. Oliveira *et al.* [12] set out several possible reasons for this failure to match predicted levels of time compensation, including duration of treatment, age, direction of shift, incomplete compensation for the sun's movement and conflict with compass information provided by other cues [12]. With our current data, we cannot rule out full or partial (imperfect) compensation and future studies should attempt to address this, for example by using controls kept at prevailing photoperiods and under identical artificial lighting to the clock-shifted hoverflies. Regardless, our comparisons of clock-shifted, rectified clock-shifted and back-transformed clock-shifted experiments with the sun compass experiment support the use of a time compensation mechanism in migrating hoverflies, a finding further supported by the lack of difference in hourly tracks and headings of radar detected hoverflies throughout the day [6,34].

Our results show that *Scaeva* spp. hoverflies use a time-compensated sun compass in their autumn migration and, together with data from radar studies in other hoverfly species, we suggest that the time-compensated sun compass may be used by the numerous other species of hoverflies found co-migrating through this pass and at other sites around the globe [6,24,26,29,47,48]. In addition, we observed many other diurnal insects co-migrating with hoverflies over the study site, including numerous species of non-syrphid Diptera as well as Lepidoptera, Hymenoptera and Odonata [29]. Are all these organisms also using a time-compensated sun compass for migration? The presence of a sun compass used for a variety of functions, and in a wide range of organisms, suggests it can be incorporated rapidly as part of the migratory syndrome [49]. To the best of our knowledge, our results are the first to confirm the use of a time-compensated sun compass for migration in insects, outside of a select few lepidopteran species, and therefore provide support for the hypothesis of repeated independent co-option of the sun compass and time compensation mechanism during the evolution of migration.

In the field, we observed a pattern of surges of migratory activity during sunny periods, with little activity when the sun was obscured. This observation, together with the loss of group orientation seen in our restricted view experiment, suggest that the sun acts as the primary celestial cue for orientation and that hoverflies wait for favourable navigational conditions, namely a visible sun, to migrate rather than relying on other cues. However, individuals flown with a restricted view are still able to maintain directed flights, as indicated by high individual *r*-values, but these are arbitrary

with respect to the celestial cues and may represent an ancestral mechanism in insects for covering large distances [50,51]. To further investigate this, future studies should determine the contribution, if any, of other celestial cues attenuated in our sun compass experiments such as intensity gradients, chromatic gradients or the polarization of light for orientation, as these may work in concert with the sun, despite being insufficient by themselves for group orientation in our experiments. Celestial polarization for example has been shown to be used by *Drosophila*, dung beetles and locusts to orientate to a vector for efficient dispersal [52–54], and by central place foragers such as desert ants and honeybees [55,56]. Foraging hoverflies have been shown to systematically circle capitula flowers while foraging, including when the sun is obscured, suggesting the ability to use not only the sun's position but also the polarization of light as orientation cues [57]. However, the polarization of light appears not to be used for orientation in migrating monarch butterflies [46], but see [58]. Finally, how hoverflies come to locate the west-southwest route through the pass of Bujaruelo during headwinds remains unclear, leaving the nature of interactions between visual cues, wind and the time-compensated sun compass to be investigated.

**Ethics.** Permission to conduct experiments was obtained from the Parc national des Pyrénées (France, authorization numbers: 2018-9 and 2019-67) and the Gobierno de Aragon (Spain, authorization numbers: 500201/24/2018/06141 and 500201/24/2019/02174).

**Data accessibility.** All data are provided as electronic supplementary material [59].

**Authors' contributions.** R.M.: conceptualization, data curation, formal analysis, investigation, methodology, resources, software, visualization, writing—original draft, writing—review and editing; W.H.: investigation, methodology, writing—review and editing; T.D.: investigation, methodology, writing—review and editing; J.T.: resources, software, writing—review and editing; M.H.M.M.: methodology, writing—review and editing; N.W.R.: methodology, writing—review and editing; J.C.: methodology, writing—review and editing; K.R.W.: conceptualization, data curation, formal analysis, funding acquisition, investigation, methodology, project administration, resources, software, supervision, validation, visualization, writing—original draft, writing—review and editing. All authors gave final approval for publication and agreed to be held accountable for the work performed therein.

**Competing interests.** We declare we have no competing interests.

**Funding.** This work was supported through grants to K.R.W. from the Royal Society University Research Fellowship scheme (grant no. UF150126). R.M. was supported through the NERC GW4+ Doctoral Training Partnership. T.D. and W.H. were supported by awards to K.R.W. from the Royal Society: a fellows enhancement award (grant no. RGF\EA\180083) and a research grant for research fellows (grant no. RGF\R1\180047), respectively. Support to M.H.M.M. was through the European Union's Horizon 2020 research and innovation programme under the Marie Skłodowska-Curie grant agreement no. 795568. N.W.R. was supported by grants from the American Airforce Research Laboratory (AFRL) (grant no. FA9550-19-1-7005) and the Bristol Centre for Agricultural Innovation (BCAI).

**Acknowledgements.** We would like to thank Michael H Dickinson for hosting K.R.W. at the California Institute of Technology during a difficult political period following his arrival on the 8 November 2016 and members of his laboratory, particularly Ysabel Milton Giraldo & Karena Cai for sharing knowledge and equipment to prototype the first hoverfly flight simulators. We would also like to thank Rochelle Meah for providing information about the diffuser and its polarization properties and Eric Postma for statistics advice. Marco Thoma assisted with the organization of the preliminary fieldwork in Switzerland, and the Commune de Champéry provided access to the accommodation at Col de Cou and Teddy Walliker provided additional assistance in the field. Finally, we would like to thank the anonymous reviewers and editor for their excellent suggestions that helped to improve our study.

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
