## [Peer Review File · Proceedings of the Royal Society B: Biological Sciences]

Review History

RSPB-2021-0936.R0 (Original submission)

Review form: Reviewer 1

Recommendation

Major revision is needed (please make suggestions in comments)

Scientific importance: Is the manuscript an original and important contribution to its field?

Good

General interest: Is the paper of sufficient general interest?

Good

Quality of the paper: Is the overall quality of the paper suitable?

Good

Is the length of the paper justified?

Yes

Should the paper be seen by a specialist statistical reviewer?

No

Do you have any concerns about statistical analyses in this paper? If so, please specify them explicitly in your report.

No

It is a condition of publication that authors make their supporting data, code and materials available - either as supplementary material or hosted in an external repository. Please rate, if applicable, the supporting data on the following criteria.

Is it accessible?

Yes

Is it clear?

Yes

Is it adequate?

Yes

Do you have any ethical concerns with this paper?

No

Comments to the Author

Please find my review in the attached pdf file.

Review form: Reviewer 2

Recommendation

Major revision is needed (please make suggestions in comments)

Scientific importance: Is the manuscript an original and important contribution to its field?

Good

General interest: Is the paper of sufficient general interest?

Good

Quality of the paper: Is the overall quality of the paper suitable?

Good

Is the length of the paper justified?

Yes

Should the paper be seen by a specialist statistical reviewer?

No

Do you have any concerns about statistical analyses in this paper? If so, please specify them explicitly in your report.

No

It is a condition of publication that authors make their supporting data, code and materials available - either as supplementary material or hosted in an external repository. Please rate, if applicable, the supporting data on the following criteria.

Is it accessible?

Yes

Is it clear?

Yes

Is it adequate?

Yes

Do you have any ethical concerns with this paper?

No

Comments to the Author

In the paper “hoverflies use a time-compensated sun compass to orientate during migration” by Massy et al., the authors performed behavioral experiments to study if hoverflies employ a time-compensated sun compass during their migration. To study this, the authors tethered individual hoverflies at the center of a flight simulator setup and allowed them to set the desired direction with respect to the sky. Without any manipulation, the hoverflies maintained highly directed flights in the expected southerly direction. However, when the authors clock-shifted a population of hoverflies by 6 hours (they delayed the perceived beginning of the day) and tested them in their flight simulator, the hoverflies kept heading directions with respect to the sky that roughly matched the perceived time of day (morning) rather than the actual time of day (afternoon). This suggests that the hoverflies use a time-compensated sky compass for orientation. In addition, the authors generated a model that allowed them to calculate how efficiently the predicted southward direction was kept by the clock-shifted hoverflies.

The behavioral data shown in Fig. 2 are solid and suggest that hoverflies rely on a time-compensated sky compass for orientation. I found the paper an easy read but have difficulties with some points. Please, see below my comments. I hope they will help to improve the paper/ to clarify some misunderstandings.

Major points:

- In general, I have difficulties with the term “partially” time-compensated compass. Why should an animal only “partially” compensate for the position of the sun? In my opinion, this does not make full sense from a biological point of view, but I am happy to discuss this with the authors in a potential revised version of the manuscript.

- In my opinion, the authors over-interpret the 70° difference between the direction taken by animals without any manipulation and the clock-shifted animals. The aspect that the hoverflies changed their heading by about 70° (rather than e.g. the expected 90° for a linear model) could, for instance, have methodical reasons, such as the tethering or the way they were kept under artificial light. I think an elegant control would have been to keep hoverflies under the same conditions (artificial light) without any clock shift. In addition, the 70° could be based on a cue conflict situation that the authors created through the time shift (e.g. sun position vs. the earth magnetic field). Given that the data show a high variance, I would be more careful with the interpretation of the change in heading. For instance, what would happen if the authors recalculated the headings in fig. 2b according to the perceived time of the clock-shifted animals? Does it look like the plot in 2a? Are the data significantly different from the data in 2a? Again, I am happy to discuss this in a potential revised version of the manuscript but in my opinion, the mean direction, combined with a high CI, is not very meaningful. I believe an interpretation of the changes in direction would require additional experiments (e.g. shifting the animals by -3h and/or advancing their circadian clock by 3/6 hours and testing the changes in heading).

- I believe the model requires some more explanation in the results. I find it difficult to understand what exactly the authors would like to show with the model. What does the efficiency of 0.6 mean? To me, it seems as if this would be efficient to roughly migrate southwards if an animal does not have a specific migration destination. What would the difference between 0.6, 0.98, and 1 efficiency mean in distance per day? In addition, why did the authors not use the actual 188.2° as their expected direction for the model?

- This might be semantic but why do the authors call it a sun compass? The hoverflies could for instance use the polarization pattern as their main orientation reference. Given that the dominant skylight cue is not known in hoverflies, I would refer to it as a “time-compensated sky compass”.

Minor points:

- Line 71: "is has been" should be "it has been"
- Lines 143-147: to the best of my knowledge, the LEE quarter white diffuser effectively cuts out all the polarized light in the UV range. Are the hoverfly's polarization detectors UV sensitive (as in other dipterans)?
- Line 184: Sun's azimuth should be sun's azimuth
- Line 227: "being the azimuth" should be "being the sun's azimuth"?
- Line 450: A reference regarding polarization orientation in locusts is missing.

Decision letter (RSPB-2021-0936.R0)

20-May-2021

Dear Dr Wotton:

I am writing to inform you that your manuscript RSPB-2021-0936 entitled "Hoverflies use a time-compensated sun compass to orientate during autumn migration" has, in its current form, been rejected for publication in Proceedings B.

This action has been taken on the advice of referees, who have recommended that substantial revisions are necessary. With this in mind we would be happy to consider a resubmission, provided the comments of the referees are fully addressed. However please note that this is not a provisional acceptance.

Sincerely,

Dr Locke Rowe

Associate Editor

Board Member: 1

Comments to Author:

Two reviewers have seen your interesting manuscript and as you will see both are very positive. However, both also have a number of substantial questions and comments, and both suggest possible new experiments to tighten the conclusions of the paper. For instance, reviewer 1 would like to have seen a control to establish that the sun is indeed used as a compass (e.g. by simulating overcast conditions or by shifting the position of an artificial sun). Both were also a bit worried about the possibility of cue conflicts. One thing that occurred to me that might go a long way to satisfying Reviewer 1 is that you have flown (non-time-shifted) hoverflies from morning to afternoon under the natural sun and have already quite a number of flight tracks. You could compare the mean direction of a population of hoverflies flown between (say) 10:30 and 11:30 with the mean direction of a population of hoverflies flown between (say) 15:30 and 16:30 (when the sun has traversed across the sky). If the mean directions are statistically indistinguishable then this would be very strong evidence for the use of the sun (or its polarisation pattern, or both) as a time-compensated compass. From your data I suspect this will be the case. At any rate, prior to publication, the comments and criticisms of the two reviewers should be adequately addressed.

Reviewer(s)' Comments to Author:

Referee: 1

Comments to the Author(s)

Please find my review in the attached pdf file.

Referee: 2

Comments to the Author(s)

In the paper "hoverflies use a time-compensated sun compass to orientate during migration" by Massy et al., the authors performed behavioral experiments to study if hoverflies employ a time-compensated sun compass during their migration. To study this, the authors tethered individual hoverflies at the center of a flight simulator setup and allowed them to set the desired direction with respect to the sky. Without any manipulation, the hoverflies maintained highly directed flights in the expected southerly direction. However, when the authors clock-shifted a population of hoverflies by 6 hours (they delayed the perceived beginning of the day) and tested them in their flight simulator, the hoverflies kept heading directions with respect to the sky that roughly matched the perceived time of day (morning) rather than the actual time of day (afternoon). This suggests that the hoverflies use a time-compensated sky compass for orientation. In addition, the authors generated a model that allowed them to calculate how efficiently the predicted southward direction was kept by the clock-shifted hoverflies.

The behavioral data shown in Fig. 2 are solid and suggest that hoverflies rely on a time-compensated sky compass for orientation. I found the paper an easy read but have difficulties with some points. Please, see below my comments. I hope they will help to improve the paper/ to clarify some misunderstandings.

Major points:

- In general, I have difficulties with the term "partially" time-compensated compass. Why should an animal only "partially" compensate for the position of the sun? In my opinion, this does not make full sense from a biological point of view, but I am happy to discuss this with the authors in a potential revised version of the manuscript.

- In my opinion, the authors over-interpret the 70° difference between the direction taken by animals without any manipulation and the clock-shifted animals. The aspect that the hoverflies changed their heading by about 70° (rather than e.g. the expected 90° for a linear model) could, for instance, have methodical reasons, such as the tethering or the way they were kept under artificial light. I think an elegant control would have been to keep hoverflies under the same conditions (artificial light) without any clock shift. In addition, the 70° could be based on a cue conflict situation that the authors created through the time shift (e.g. sun position vs. the earth magnetic field). Given that the data show a high variance, I would be more careful with the interpretation of the change in heading. For instance, what would happen if the authors

recalculated the headings in fig. 2b according to the perceived time of the clock-shifted animals? Does it look like the plot in 2a? Are the data significantly different from the data in 2a? Again, I am happy to discuss this in a potential revised version of the manuscript but in my opinion, the mean direction, combined with a high CI, is not very meaningful. I believe an interpretation of the changes in direction would require additional experiments (e.g. shifting the animals by -3h and/or advancing their circadian clock by 3/6 hours and testing the changes in heading).

- I believe the model requires some more explanation in the results. I find it difficult to understand what exactly the authors would like to show with the model. What does the efficiency of 0.6 mean? To me, it seems as if this would be efficient to roughly migrate southwards if an animal does not have a specific migration destination. What would the difference between 0.6, 0.98, and 1 efficiency mean in distance per day? In addition, why did the authors not use the actual 188.2° as their expected direction for the model?

- This might be semantic but why do the authors call it a sun compass? The hoverflies could for instance use the polarization pattern as their main orientation reference. Given that the dominant skylight cue is not known in hoverflies, I would refer to it as a "time-compensated sky compass".

Minor points:

- Line 71: "is has been" should be "it has been"

- Lines 143-147: to the best of my knowledge, the LEE quarter white diffuser effectively cuts out all the polarized light in the UV range. Are the hoverfly's polarization detectors UV sensitive (as in other dipterans)?

- Line 184: Sun's azimuth should be sun's azimuth

- Line 227: "being the azimuth" should be "being the sun's azimuth"?

- Line 450: A reference regarding polarization orientation in locusts is missing.

Author's Response to Decision Letter for (RSPB-2021-0936.R0)

See Appendix A.

RSPB-2021-1805.R0

Review form: Reviewer 1

Recommendation

Accept as is

Scientific importance: Is the manuscript an original and important contribution to its field?

Good

General interest: Is the paper of sufficient general interest?

Good

Quality of the paper: Is the overall quality of the paper suitable?

Good

Is the length of the paper justified?

Yes

Should the paper be seen by a specialist statistical reviewer?

No

Do you have any concerns about statistical analyses in this paper? If so, please specify them explicitly in your report.

No

It is a condition of publication that authors make their supporting data, code and materials available - either as supplementary material or hosted in an external repository. Please rate, if applicable, the supporting data on the following criteria.

Is it accessible?

N/A

Is it clear?

N/A

Is it adequate?

N/A

Do you have any ethical concerns with this paper?

No

Comments to the Author

Well done, nice manuscripts.

Decision letter (RSPB-2021-1805.R0)

27-Aug-2021

Dear Dr Wotton

I am pleased to inform you that your Review manuscript RSPB-2021-1805 entitled "Hoverflies use a time-compensated sun compass to orientate during autumn migration" has been accepted for publication in Proceedings B.

The referee(s) do not recommend any further changes. Therefore, please proof-read your manuscript carefully and upload your final files for publication. Because the schedule for publication is very tight, it is a condition of publication that you submit the revised version of your manuscript within 7 days. If you do not think you will be able to meet this date please let me know immediately.

To upload your manuscript, log into <http://mc.manuscriptcentral.com/prsb> and enter your Author Centre, where you will find your manuscript title listed under "Manuscripts with Decisions." Under "Actions," click on "Create a Revision." Your manuscript number has been appended to denote a revision.

You will be unable to make your revisions on the originally submitted version of the manuscript. Instead, upload a new version through your Author Centre.

- 1) A text file of the manuscript (doc, txt, rtf or tex), including the references, tables (including captions) and figure captions. Please remove any tracked changes from the text before submission. PDF files are not an accepted format for the "Main Document".

2) A separate electronic file of each figure (tiff, EPS or print-quality PDF preferred). The format should be produced directly from original creation package, or original software format. Please note that PowerPoint files are not accepted.

3) Electronic supplementary material: this should be contained in a separate file from the main text and the file name should contain the author's name and journal name, e.g. `authorname_procb_ESM_figures.pdf`

All supplementary materials accompanying an accepted article will be treated as in their final form. They will be published alongside the paper on the journal website and posted on the online figshare repository. Files on figshare will be made available approximately one week before the accompanying article so that the supplementary material can be attributed a unique DOI. Please see: <https://royalsociety.org/journals/authors/author-guidelines/>

4) Data-Sharing and data citation

It is a condition of publication that data supporting your paper are made available. Data should be made available either in the electronic supplementary material or through an appropriate repository. Details of how to access data should be included in your paper. Please see <https://royalsociety.org/journals/ethics-policies/data-sharing-mining/> for more details.

<http://datadryad.org/submit?journalID=RSPB&manu=RSPB-2021-1805> which will take you to your unique entry in the Dryad repository.

Once again, thank you for submitting your manuscript to Proceedings B and I look forward to receiving your final version. If you have any questions at all, please do not hesitate to get in touch.

Sincerely,

Dr Locke Rowe

Associate Editor

Comments to Author:

Reviewer 1 is now fully satisfied with your revisions to address their comments, and I am also very satisfied with the extra analyses you have done and the new data you have added, which I agree shows quite convincingly that hover flies use a sun compass for navigation. This is an important and interesting addition to the insect navigation literature. Thanks for sending your manuscript to Proceedings B!

Reviewer(s)' Comments to Author:

Referee: 1

Comments to the Author(s).

Well done, nice manuscripts.

Sincerely,

Proceedings B

Decision letter (RSPB-2021-1805.R1)

31-Aug-2021

Dear Dr Wotton

I am pleased to inform you that your manuscript entitled "Hoverflies use a time-compensated sun compass to orientate during autumn migration" has been accepted for publication in Proceedings B.

If you are likely to be away from e-mail contact please let us know. Due to rapid publication and an extremely tight schedule, if comments are not received, we may publish the paper as it stands. If you have any queries regarding the production of your final article or the publication date please contact procb_proofs@royalsociety.org

Data Accessibility section

Open Access

Paper charges

Sincerely,

Proceedings B

Appendix A

Centre for Ecology and Conservation
University of Exeter
Penryn Campus,
Penryn, Cornwall
TR10 9FE, UK
e: k.r.wotton@exeter.ac.uk
w: www.exeter.ac.uk

11 August 2021

Response to referees: manuscript RSPB-2021-0936 entitled "Hoverflies use a time-compensated sun compass to orientate during autumn migration"

Dear editor,

We would like to thank both you and the reviewers for the helpful comments. We believe we have been able to address all of these and below we set out our responses in blue together with a summary of the comments addressed.

In brief, we present additional evidence that (1) migratory hoverflies undertake directed flight and that this group orientation increases in flights undertaken later in the day; (2) that hoverflies use the sun as their primary cue; and (3) that our hoverflies are indeed clock-shifted.

However, this work has substantially increased the length of our paper beyond the 10-page limit of *Proceedings B* and we have therefore chosen to remove the model describing the efficiency of southwards migration under varying degrees of time-compensation. It is our intention to resubmit this work as a standalone paper to *Biology Letters* in due course.

Despite this, our much improved paper remains the first confirmation of the use of a time-compensated sun compass for migration in insects outside of a select few lepidopteran species, and therefore provides strong support for its repeated co-option, hypothesised to have occurred during the evolution of migration in different lineages.

I hope you agree that this update has substantially enhanced the manuscript and we look forward to your response.

Yours sincerely,

Karl Wotton

Associate Editor Board Member

Editor comment: One thing that occurred to me that might go a long way to satisfying Reviewer 1 is that you have flown (non-time-shifted) hoverflies from morning to afternoon under the natural sun and have already quite a number of flight tracks. You could compare the mean direction of a population of hoverflies flown between (say) 10:30 and 11:30 with the mean direction of a

population of hoverflies flown between (say) 15:30 and 16:30 (when the sun has traversed across the sky). If the mean directions are statistically indistinguishable then this would be very strong evidence for the use of the sun (or its polarisation pattern, or both) as a time-compensated compass. From your data I suspect this will be the case.

Our reply: We thank the editor for this excellent suggestion that led us to explore deeper into the time-associated variation in our data. Unfortunately, the suggested analysis was not possible as we have only 3 datapoints for each of these time windows (morning: 224.8°, 271.4°, 99.8° and evening: 222.2°, 230.3°, 214.8°) and a Mardia-Watson-Wheeler test requires at least 10 data points. We attempted instead to run the comparison on the first half (10:37 to 13:53) and second half (13:57 up to 16:34) of the data. We found that the first 15 data points show no group orientation, while the 15 flights after 13:57 show a strong group mean direction. We interpret this finding using new data on the diurnal migration activity and temperature observed on the mountain pass (see text below and Figure 2 in manuscript). This new analysis indicates that the relatively low r - and p -values for the group mean using the whole data set were driven by these earlier flights.

New manuscript text: Hoverflies flown in the sun-compass experiment, with the sun visible, landscape cues obscured, and other celestial cues attenuated by photographic diffuser, headed almost due south (Moore's modified Rayleigh test, $\theta = 188.2^\circ$, $n = 30$, $R^* = 1.294$, $p < 0.01$; Figure 2a). Diurnal activity patterns of migrants crossing the mountain pass show a marked increase beginning after 14:00, coinciding with increasing temperatures (Figure 2b). Splitting the data of the sun compass experiment in half, hoverflies flown before 13:57 orientated randomly (Moore's modified Rayleigh test, $n = 15$, $R^* = 0.375$, $p < 0.9$; Figure 2c) whereas hoverflies flown after this time, hereafter referred to as the 'last 15', displayed a more directed group heading to the south-southwest (Moore's modified Rayleigh test, $\theta = 194.4^\circ$, $n = 15$, $R^* = 1.480$, $p < 0.0001$; Figure 2d).

Our reply continued: We also provide additional data, including back-transformed clock-shift data, as suggested by referee 2, and analysis of the last 15 flown hoverflies that provides stronger support for the clock-shift results (see text below).

New manuscript text: As the clock-shifted experiments were undertaken later in the day when the sun would naturally be in a more clockwise position, to rule out simple phototaxis, the analyses were repeated on angles that were rectified to be relative to the direction of the sun, so that the direction of the sun is in the 180° position (Figure S1). The rectified mean directions of hoverflies under normal and clock-shifted conditions still differed significantly, as confirmed by a Mardia-Watson-Wheeler test (all: $W = 7.75, p = 0.021$; last 15: $W = 10.918, p = 0.0043$), while back transforming the angle of the clock-shifted hoverflies based on their perceived time results in a group heading that is not significantly different from the sun compass (all: $W = 2.76, p = 0.251$; last 15: $W = 4.06, p = 0.132$). Together this indicates that the hoverflies were clock-shifted, flying in different directions with respect to the sun (Figure 3a-c) and compensating for its position rather than just following its course throughout the day.

Our reply continued: As a final test of directional differences, we took the last 20 flown hoverflies and divided them into 2 groups of 10 to allow a Mardia-Watson-Wheeler test to be carried out, while also reducing earlier datapoints that lacked a group orientation. The Mardia-Watson-Wheeler test confirmed that the mean directions in the two samples were not significantly different (Figure below; $W = 1.65, p = 0.44$), indicating that group orientation did not change despite the sun traversing approximately 63° through the sky. However, we have chosen not to include this data as the mean orientation of the first 10 hoverflies is not significant ($n = 10, r = 0.17, p = 0.69$).

Figure: Flight directions of hoverflies flown at 13:11-14:26 (black) and 14:34-1634 (red).

Referee: 1

Referee comment: the authors rectified the dataset by shifting the recorded orientation angles by the difference between the azimuth of the sun and “180 degrees” (it is unclear if this means geographic South).

Our reply: We have now clarified that 180 degrees means geographic south

Referee comment: Unfortunately, the manuscript at its' current stage lacks the critical control experiment which would demonstrate the usage of the sun as a compass cue. Neither did the authors experimentally shift the position of the sun (by e.g. the use of mirrors), nor did they systematically obscure the view of the sun (recordings done under e.g. a simulated overcast condition)

Our reply: As suggested by the reviewer we have now included a control experiment from previously unanalysed data where the view of the sun was obscured by fitting a smaller restrictor ring to the top of the flight simulator. The restrictor presents a 70° field of view to the hoverfly that excludes the sun but includes intensity gradients, chromatic gradients and polarized light cues. Hoverflies flown under these conditions orientated randomly (Moore's Rayleigh test $n = 42$, $R^* = 0.634$, $p < 0.5$). In addition, to account for the ambiguous nature of the polarized light cue, where opposite directions along an axis cannot be distinguished, we tested for a bidirectional line-up response by bidirectionally transforming the angles and repeated the test, however the results remained non-significant ($n = 42$, $R^* = 0.585$, $p < 0.5$). We believe this provides the necessary control to evidence for the use of the sun as a compass cue and have updated the text and Figure 2 accordingly.

Referee comment: I would like to encourage the authors to separate the orientation data of the two species and plot two circular diagrams (like in Fig.2) for *Scaeva pyrastris* and *S. Selenitica* respectively (to allow a comparison between the two species). Please do so for the clock-shifted animals as well.

Our reply: We have updated figure 2 with this information.

Referee comment: The authors mention in line 143 that they used diffuser paper to cover the opening of the arena, minimizing a light artifact in the arena. So the animals could see a somewhat distorted image of the sky? If so, please mention this in line 247/248.

Our reply: We have clarified this. The sentence now reads: “Hoverflies flown in the sun-compass experiment, with the sun visible, landscape cues obscured, and other celestial cues attenuated by photographic diffuser, headed almost due south (Moore’s modified Rayleigh test, $\theta = 188.2^\circ$, $n = 30$, $R^* = 1.294$, $p < 0.01$; Figure 2a)”.

Referee comment: How often was an individual tested? - Were all animals caught at the same time and then divided into two groups, one being clockshifted and the other one not being clock-shifted?

Our reply: We have clarified this in the text. A separate batch of flies were caught and subjected to a clock-shift experiment while sun compass flies were reused in the restricted view experiment. Individuals were only tested once in each experiment.

Referee comment: When were the clock-shift experiments conducted relative to the “normal” orientation experiments? Were the experiments conducted e.g. on two following weeks or intermingled with each other (e.g. normal experiments until early afternoon and then clock-shift experiments in the late afternoon)?

Our reply: We have clarified this in the text, and now state that time-compensation “Experiments were carried out between the 7th and 11th of October 2019 between 14:29 and 16:42, overlapping with a subset of the sun compass and restrictor experiments.”

Referee comment: Did you ever test the same individuals at 2 different times of the day? If yes, did you detect a change in the orientation angles? - Was the arena rotated between trials?

Our reply: We did not carry out these experiments but agree that that would be a worthwhile follow up if, covid allowing, we are able to return to the field at some point in the future.

Referee: 2

Referee comment: In general, I have difficulties with the term “partially” time-compensated compass. Why should an animal only “partially” compensate for the position of the sun? In my opinion, this does not make full sense from a biological point of view, but I am happy to discuss this with the authors in a potential revised version of the manuscript.

Our reply: The idea that the sun compass may be fully or only partially time-compensated is not ours and has been discussed in the literature by several authors (see Oliveira et al. 1998 *Journal of Experimental Biology*; Srygley, Robert & Dudley 2008 *Integrative and Comparative Biology*; or as degrees of compensation here: Guilford & Taylor 2014 *Animal Behaviour*). Partial compensation has been seen in several groups (crustacean sandhoppers: Wallraff, 1981; honeybees: Gould, 1980; pigeons: Schmidt-Koenig et al. 1991a; Wiltschko et al. 1994; Chappell, 1997), for example desert ants underestimate the rate of movement of the sun's azimuth when it is high and overestimate this rate of movement when it is low (Wehner, 1984). We have attempted to clarify this in the manuscript (see comments below).

Referee comment: In my opinion, the authors over-interpret the 70° difference between the direction taken by animals without any manipulation and the clock-shifted animals. The aspect that the hoverflies changed their heading by about 70° (rather than e.g. the expected 90° for a linear model) could, for instance, have methodical reasons, such as the tethering or the way they were kept under artificial light. I think an elegant control would have been to keep hoverflies under the same conditions (artificial light) without any clock shift.

Our reply: We agree with the reviewer and have clarified this at various points in the text. We now state in the results: “The mean direction of the time-shift experiment was 68.9° clockwise from that of the sun-compass experiment, representing 70% of the 98.1° adjustment required to fully compensate for the mean six-hour azimuth change, although the confidence interval does not rule out complete time compensation.” In addition, we have added new clarifying text to the discussion:

New manuscript text: As has been demonstrated in migrating butterflies [12,13], our time-shift experiment shows that migratory hoverflies compensate for the position of the sun as it changes throughout the day. The degree of compensation observed in our experiments was approximately two thirds of the average change in azimuth, although confidence intervals do not rule out complete compensation as compared to the mean direction of the sun compass experiment. Similar results have been documented for neotropical migrant butterflies while orientations of monarch butterflies more closely match predicted shifts [12,13]. Oliveira, Srygley & Dudley (1998) set out several possible

reasons for this failure to match predicted levels of time-compensation, including duration of treatment, age, direction of shift, incomplete compensation for the sun's movement and conflict with compass information provided by other cues. With our current data we cannot rule out full or partial (imperfect) compensation and future studies should attempt to address this, for example by utilising controls kept at prevailing photoperiods and under identical artificial lighting to the clock-shifted hoverflies. Regardless, our comparisons of clock-shifted, rectified clock-shifted and back-transformed clock-shifted experiments with the sun compass support the use of a time-compensation mechanism in migrating hoverflies, a finding also supported by the lack difference between mean directions in hourly tracks and headings of radar detected hoverflies throughout the entire day.

Referee comment: In addition, the 70° could be based on a cue conflict situation that the authors created through the time shift (e.g. sun position vs. the earth magnetic field). Given that the data show a high variance, I would be more careful with the interpretation of the change in heading. For instance, what would happen if the authors recalculated the headings in fig. 2b according to the perceived time of the clock-shifted animals? Does it look like the plot in 2a? Are the data significantly different from the data in 2a? Again, I am happy to discuss this in a potential revised version of the manuscript but in my opinion, the mean direction, combined with a high CI, is not very meaningful. I believe an interpretation of the changes in direction would require additional experiments (e.g. shifting the animals by -3h and/or advancing their circadian clock by 3/6 hours and testing the changes in heading).

Our reply: We thank the referee for bringing up this interesting possibility. We have investigated this further in relation to the magnetic tethering system used in our experiments. In our system the hoverfly is exposed to a symmetrical magnetic field of more than 400 gauss (<https://www.kjmagnetics.com/magfield.asp?pName=RX8CC>). We believe this makes it highly unlikely that the weak magnetic field emanating from the Earth's surface of 0.25 to 0.65 gauss would be able to provide a suitable conflicting cue. In addition, we have carried out the suggested back transformation of the clock-shifted data and show that the sun compass and back transformed data are not significantly different, supporting our initial interpretation that the hoverflies are clock-shifted. The modified results section is copied below.

New manuscript text: As the clock-shifted experiments were undertaken later in the day when the sun would naturally be in a more clockwise position, to rule out simple phototaxis, the analyses were repeated on angles that were rectified to be relative to the direction of the sun, so that the direction of the sun is in the 180° position. (Figure S1). The rectified mean directions of hoverflies under normal and clock-shifted conditions still differed significantly, as confirmed by a Mardia-Watson-Wheeler test (all: $W = 7.75$, $p = 0.021$; last 15: $W = 10.918$, $p = 0.0043$), while back transforming the angle of the clock-shifted hoverflies based on their perceived time results in a group heading that is not significantly different from the sun compass (all: $W = 2.76$, $p = 0.251$; last 15: $W = 4.06$, $p = 0.132$), together this indicates that the hoverflies were clock-shifted, flying in different directions with respect to the sun (Figure 3a-c) and compensating for its position rather than just following its course throughout the day.

Our reply continued: As suggested we have tried to provide a more careful interpretation of our findings, discuss potential sources of error and suggest follow up experiments that may help to clarify the findings (see new text presented in response to previous comment). Regarding the high confidence intervals, we agree that they are not ideal, however we disagree that combined with the mean direction that they are not very meaningful. The mean directions are statistically significant as confirmed by a Mardia-Watson-Wheeler test, and this significance is increased when analysing the last half of the data set (p-value changes from 0.02 to 0.0089) and the rectified last half of the data set (p-value from 0.02 to 0.0043). In addition, the confidence intervals are reduced in the analysis of the last half of the data set from 71° to 53°, approaching the 36° seen by Mouritsen & Frost for the monarch butterfly. We believe that this, together with the new analysis of the back transformed clock-shift data, indicates that the hoverflies are indeed clock-shifted, were flying in different directions with respect to the sun and compensating for its position rather than just following its course throughout the day. Finally, while additional clock-shift experiments would be a nice addition and may help, with the right set up, to distinguish between full, partial, and time-averaging

methods, we do not believe that it is necessary to demonstrate that the sun compass is time-compensated as was our goal here.

Referee comment: I believe the model requires some more explanation in the results. I find it difficult to understand what exactly the authors would like to show with the model. What does the efficiency of 0.6 mean? To me, it seems as if this would be efficient to roughly migrate southwards if an animal does not have a specific migration destination. What would the difference between 0.6, 0.98, and 1 efficiency mean in distance per day? In addition, why did the authors not use the actual 188.2° as their expected direction for the model?

Our reply: We attempted to state what we would like to show in the text of the results, however, on re-reading we agree that this may have been too vague. We have now chosen to remove these sections due to space constraints caused by our additional analyses that has pushed us far over the 10-page limit set by *ProcB*. Regardless, are glad to have receive this feedback on various aspects of our model and have addressed each issue raised for a standalone manuscript to be submitted in due course.

Referee comment: This might be semantic but why do the authors call it a sun compass? The hoverflies could for instance use the polarization pattern as their main orientation reference. Given that the dominant skylight cue is not known in hoverflies, I would refer to it as a “time-compensated sky compass”.

Our reply: We agree with the referee on this point given the evidence we provided in the original version. However, our new manuscript includes a ‘restricted view’ experiment that demonstrates loss of group orientation when the sun is obscured but other cues such as intensity gradients, chromatic gradients and polarization of light remain. We believe this clearly demonstrates that the sun is the dominant skylight cue and therefore the term ‘sun compass’ is now accurate.

Referee comment: Line 71: “is has been” should be “it has been”

Our reply: Done

Referee comment: Lines 143-147: to the best of my knowledge, the LEE quarter white diffuser effectively cuts out all the polarized light in the UV range. Are the hoverfly's polarization detectors UV sensitive (as in other dipterans)?

Our reply: Unfortunately, we did not measure light in the UV range, and we are unaware of any measures of the sensitivity of hoverfly polarization detectors in the dorsal rim area of the eye. We note however, that while this signal may have been attenuated in the sun compass experiment, it was not in the restrictor experiments that were carried out with an open lid. Therefore, while we cannot rule out a role for polarized light in combination with the sun, it is not sufficient by itself for orientation in a seasonally beneficial direction. We now cover these points in the modified text:

New manuscript text: In the field we observed a pattern of surges of migratory activity during sunny periods, with little activity when the sun was obscured. This observation, together with the loss of group orientation seen in our restricted-view experiment, suggest that the sun acts as the primary celestial cue for orientation and that hoverflies wait for favourable navigational conditions, namely a visible sun, to migrate rather than relying on other cues. However, individuals flown with a restricted view are still able to maintain directed flights, as indicated by high individual *r*-values, but these are arbitrary with respect to the celestial cues and may represent an ancestral mechanism in insects for covering large distances [51, 52]. To further investigate this, future studies should determine the contribution, if any, of other celestial cues attenuated in our sun compass experiments such as intensity gradients, chromatic gradients or the polarization of light for orientation, as these may work in concert with the sun, despite being insufficient by themselves for group orientation in our experiments.

Referee comment: Line 184: Sun's azimuth should be sun's azimuth

Our reply: Done

Referee comment: Line 227: "being the azimuth" should be "being the sun's azimuth"?

Our reply: Done

Referee comment: Line 450: A reference regarding polarization orientation in locusts is missing.

Our reply: We now cite Homberg U et al (2011) Central neural coding of sky polarization in insects.

Philos Trans R Soc Lond B Biol Sci 366:680–687